

# An interactive audio-visual installation using ubiquitous hardware and web-based software deployment

Tiago Fernandes Tavares

School of Electrical and Computer Engineering, University of Campinas, Brazil

## ABSTRACT

This paper describes an interactive audio-visual musical installation, namely MOTUS, that aims at being deployed using low-cost hardware and software. This was achieved by writing the software as a web application and using only hardware pieces that are built-in most modern personal computers. This scenario implies in specific technical restrictions, which leads to solutions combining both technical and artistic aspects of the installation. The resulting system is versatile and can be freely used from any computer with Internet access. Spontaneous feedback from the audience has shown that the provided experience is interesting and engaging, regardless of the use of minimal hardware.

## INTRODUCTION

Artistic interactive musical installations, like Aether (*Sanchez & Castro, 2014*) and Intrium (*Guisan, 2005*), are devices that allow an audience to interact with a sonic environment or musical concept using electronic sensors. In some cases, the installation is built as to augment the interaction between the public and an specific environment, as the well-known piano staircase (*TheFunTheory, 2009*), an installation in which each step in a staircase was behaved like the key of a piano, thus causing music to be played when the audio went downstairs and upstairs. More recently, modern motion sensors allowed achieving new possibilities of musical performance and interaction (*Jung et al., 2012*; *Chen, Maeda & Takahashi, 2014*) by mapping movements into musical responses.

Interactive musical devices present both artistic and technological challenges (*Garnett, 2001*). They create the possibility of generating music according to a dance, instead of constraining dance to a pre-defined musical piece (*Morales-Manzanares et al., 2001*). Hence, they bring to the public a technology-enabled experience that is perceptively different from simply listening to music or dancing to a recording.

Nevertheless, most installations are expensive artifacts that must be mounted by a well-trained team. This causes their cultural experience to be restricted to specific environments, such as art galleries, museums or particular events. Therefore, the cultural transformation derived from the interaction with a novel music concept has a limited audience range.

Corresponding author
Tiago Fernandes Tavares,
tavares@dca.fee.unicamp.br

The installation proposed in this article, namely MOTUS, aims at being deployed for a broad public. This is achieved by combining a web-deployed software stack, little hardware requirements and simple, yet engaging, methods for interaction. As a result, the experience provided by MOTUS is made accessible for any person with an internet connection and a laptop with a webcam.

The installation uses a camera as a sensor device, and a simple motion detection algorithm (*Wirayuda et al., 2013*) to characterize the audience's movements. The musical generation, based on Markov chains (*Schulze & Van der Merwe, 2011*; *Pachet, 2002*; *Cope, 1997*), aims at converting the detected movement intensity into the intensity of the musical manifestation without requiring previous musical knowledge from the audience. The installation software also comprises auditory and visual feedback, which may use the laptop's hardware (screen and speakers) or external devices such sound reinforcement systems and projectors.

The remainder of this article is organized as follows. First, related work is presented in 'Related Work,' followed by a discussion about the artistic concepts behind the development of MOTUS in 'Artistic Concept.' In 'The Installation,' MOTUS is thoroughly described both from the artistic and the technical points of view. Further discussion, based on interactions with the audience, is conducted in 'Audience Feedback and Discussion.' Last, 'Conclusion' brings conclusive remarks.

## RELATED WORK

A great number of interactive art installations has been constructed in the last decade. Each one of them implements an underlying purpose, which is often discussed in academic publications. Some are especially related to MOTUS, as it will be discussed below.

*Birchfield et al. (2006)* brought forward the question of placement of an installation, and its impact on the usage of a public space. After implementing sonification of a bus stop in a busy street, they observed that the general public often feels self-conscious about producing sounds in this environment. Henceforth, audience engagement is an important, non-trivial issue to be considered in installations.

A possible technique to achieve audience engagement is to develop a specific space for the installation, providing both auditory and visual stimuli (*Kobori et al., 2006*; *Seo & Corness, 2007*). However, as observed in the Piano Staircase (*TheFunTheory, 2009*), audience engagement may happen even if the installation is placed in a public space. This indicates that the placement of the installation does not cause audience engagement alone.

In the evaluation of the interactive dance installation Hoppsa Universum (*Kallblad et al., 2008*), it has shown that is audience perception was frequently described with expressions like *it was fun* or *be with friends*. Later, *Schacher (2009)* noted that the audience engagement is related to the fast understanding of the interaction model, which may restrict the usage of more complicated interfaces or algorithms.

*Morreale, Masu & Angeli (2013)* presented an algorithm, namely Robin, capable of generating piano music from the spatial position of members of the audience. The algorithm uses a rule-based system that models Western piano style music, and may

be used by untrained (non-musician) members of the audience. It was presented in an installation that was well-evaluated, with great acceptance ratios.

Motus considers all of these aspects, but, unlike the work discussed above, it does not require special hardware (other than that present in most current laptops) or preparations to be used. It aims at being easily used, including by untrained audience, which reflects on the simplicity of its interaction model and its software is deployed as a web application, thus it can be readily used in private spaces. Motus is thoroughly described in the next section.

## ARTISTIC CONCEPT

Motus was first idealized from the idea of converting movements to music using a camera. Its name comes from the Latin word that means "motion." This section describes the artistic concepts over which it was constructed.

The musical concept behind Motus was derived from improvised genres, like Free Jazz and some styles of ethnic Drum Circles. During an improvisation session, it is important to perceive the other members of the ensemble and create some form of communication with them. In this context, elements such as harmony and rhythm may be transformed to fit the communication process that emerges in each session.

According to the model presented by *Dubberly, Pangaro & Haque (2009)*, this type of interaction is mediated by the intention of each agent. This means that the correspondence to an intention is, for the improvisation group, more important than the achievement of technical precision. Therefore, Motus uses a music generation model that responds to the audience intention.

For the construction of the interactive system, this intention must be assigned to control a measurable aspect of the generated music. Since Motus is intended to be used by an untrained audience, the musical aspect controlled by the audience's intention must be simple to understand. For this reason, the audience's intention was assigned to control the musical intensity.

To evaluate the audience's intention using the webcam, it was necessary to estimate the intensity of captured movements. Instead of mapping particular movements to specific sonic representations, a general movement intensity was measured using pixel-by-pixel differences. This allows the audience to explore not only the interaction with Motus, but also the diverse possibilities of using their bodies, interacting with friends or using objects.

With the goal of inducing broader movements, the video area was divided into different regions, each related to a sonic representation. The audience can visualize the video feed, with a color scheme that highlights the regions that are most active. In addition to the aesthetic appeal, this feedback helps understanding the interaction process.

For this same reason, piano sounds were used for audio rendering. They have the goal of being easy to recognize, as most of the general audience (at least in Western countries) is familiar with the instrument. The installation is described from a more technical point of view in the next section.

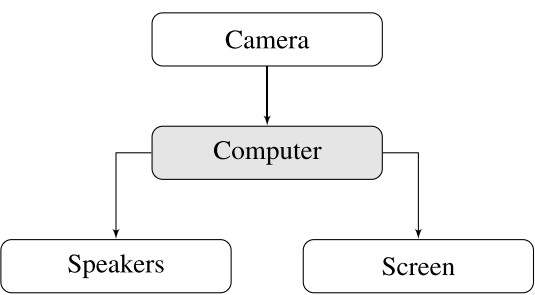

**Figure 1  Installation overview.**

## THE INSTALLATION

The main concern when developing MOTUS was that it could be used by as many people as possible. Steps towards this goal were taken by requiring as little external hardware as possible and by deploying the software as a web application. The hardware necessary to mount the installation was restricted to that available in a common laptop, i.e., a webcam, a video screen and internal speakers, leading to an overall system as described in Fig. 1.

The deployment problem can be solved by using JavaScript as the main programming language. It can be used to deploy the application directly on a host web browser. However, this choice also poses a performance restriction, as JavaScript applications are usually slow when compared to native (compiled) programs.

On the artistic side, the concept behind MOTUS is that it should convert movement to music. This conversion means that user movements should trigger a musical response, and more intense movements should correspond to a more intense musical response. Therefore, two subsystems are necessary: one comprising a movement detection algorithm and another one containing a musicological model that generates musical responses.

Also, it quickly became clear that a video feedback of the detection process could improve the audience's experience. This happens because the visual information allows the user to understand and appropriate their interaction with a novel musical device. As a result, a greater level of immersion could be provided.

Therefore, MOTUS can be detailed in a block diagram as shown in Fig. 2. All blocks in gray are software, and will be executed within the computer shown in Fig. 1. The following sub-sections will present a thorough description of the movement detection, video rendering, the musicological model and the audio rendering system.

### Movement detection

The movement detection process applied in MOTUS is very simple, as the web-based implementation does not allow for computationally demanding algorithms. The algorithm begins with the calculation the *value* $v_p$ of each pixel $p$ as the sum of its red, green and blue channels, as it is a common practice in computer vision algorithms (*Szeliski, 2010*). Hence, it may be expressed by:

$$v_p = r_p + g_p + b_p. \tag{1}$$

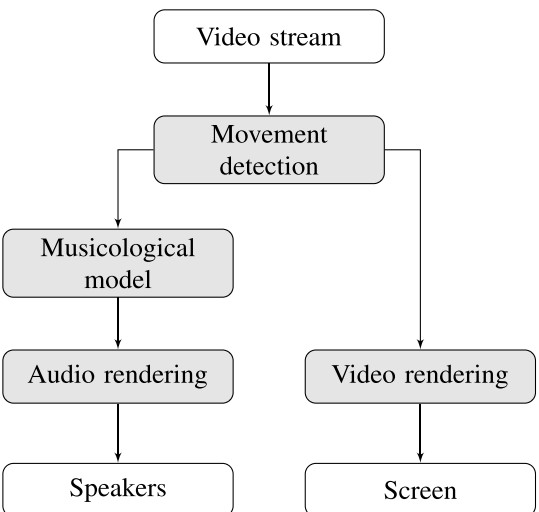

**Figure 2  Block diagram describing the installation.**

In this algorithm, it is more important to detect the intensity of movements than the precise movement location. Such a parameter can be estimated using the mean absolute difference between the pixel values in a frame $t$ and those in the previous frame $t - 1$ (*Moeslund, 2012*), that is:

$$\mu[t] = \frac{\sum_{p=1}^{P} |v_p[t] - v_p[t-1]|}{P}, \tag{2}$$

where $P$ is the number of pixels in the frame.

Calculating the amount of movement in the whole video feed, however, does not allow placing different interaction with the installation when performing different types of movements. Therefore, the video input was first split into four different partitions. Each partition had its own movements intensity estimation, and, as will be seen later, is related to a different part of the interaction experience.

In preliminary tests, it was noticed that $\mu[t]$ changes too quickly, which gives an impression of chaos and lack of control. Hence, it was necessary to apply a filter to each $\mu_t$ signal before using it for further purposes. An attack-release filter was applied, using the following expression:

$$\hat{\mu}[t] = \begin{cases} \alpha\mu[t] + (1-\alpha)\hat{\mu}[t-1] & \text{if } \mu[t] > \hat{\mu}[t-1] \\ \beta\mu[t] + (1-\beta)\hat{\mu}[t-1] & \text{if } \mu[t] \leq \hat{\mu}[t-1]. \end{cases} \tag{3}$$

The attack-release filter acts as a low-pass filter whose cut-off frequency is different whether the input signal is higher or lower than the last output. Higher values for the $\alpha$ and $\beta$ coefficients correspond to shorter attack and release times, respectivelly. They were manually adjusted so that the resulting interaction was smooth as desired.

Hence, the result of the movement detection process is a set of four movement estimates $\hat{\mu}[t]$, one for each partition. This result was used both in the musicological model and the video rendering process, as it will be discussed later.

## Video rendering

The visual feedback provided by MOTUS aims at two correlated but different goals. The first is to yield feedback on what the system is doing; that is, what is being detected. The second is to make the audience experience more immersive and engaging.

Three dimensions of the system's inner mechanisms were chosen to be conveyed: the captured image values $v_p$ as in Expression (1), the differences between the current frame and the previous frame ($|v_p[t] - v_p[t-1]|$) and the final detected movement intensity in each partition $\hat{\mu}[t]$ as in Expression (3). To allow the audience to clearly visualize each aspect of the interaction, these parameters were mapped to different colors. These colors were arbitrarily chosen to be blue, red and green, which colored the feedback video creating a particular aesthetic environment.

As stated before, the values of each frame were mapped to the blue channel of the feedback video. The blue color, then, becomes dominant at almost all times, which gives the installation a general feeling of blue. As a consequence, blue is a color related to musical rest.

Each pixel's absolute difference to the previous frame was mapped to the red channel. This caused a red "ghost" to appear in point where strong movements were detected, indicating that an interaction was detected. This piece of visual feedback is bounded to the user and became subtle when compared to other cues.

The amount of movement $\hat{\mu}[t]$ in each frame partition was mapped to the green channel of the corresponding pixels. This aimed at helping the audience to relate movements to sounds, as a particular category of sonic responses would be clearly correlated to specific blinks in a region of the screen. This piece of visual feedback is strongly correlated to the musicological model employed, as it will be seen below.

A screenshot of the video feedback in action is shown in Fig. 3, converted to gray scale to ensure visibility in printed media. As it can be seen, screen areas in which there is more movement are highlighted, and it is possible to visualize both the body movement detection and the activation of screen areas related to musical responses. Thus, the audience's impact on the audiovisual environment is easily visualized.

## Musicological model

The generation of musical sequences was done by means of four musicological models, each receiving as input the amount of movement of a different video partition. In all cases, the model should yield a musical manifestation that is perceived as more intense when movements in that partition are more intense. Also, this correlation should be perceived almost immediately.

In addition to that, the models were built so that no strong sensation of downbeat would emerge, hence avoiding inducing the audience to perform known popular dance moves and favoring the exploration of different body movements. The sensation of

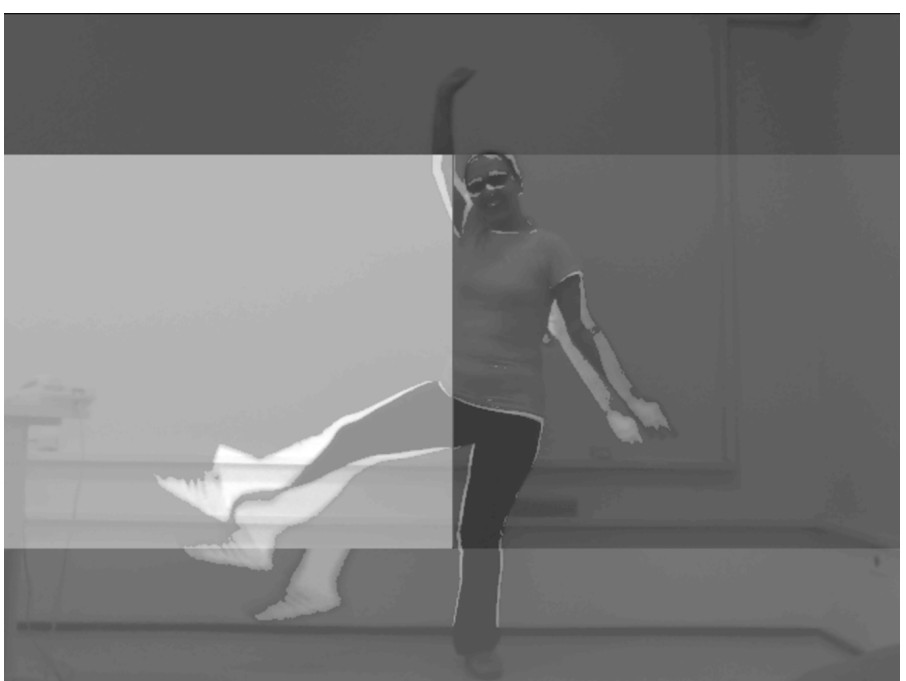

**Figure 3** Screenshot of the video render.

closure commonly found in tonal music (e.g., in I–IV–V–I sequences) was also avoided, preventing the comparison of the generated music with known pieces, also favoring experimentation. To keep the interaction more interesting, each partition was bounded to a different musicological behavior, which aimed at inducing the audience to explore the whole interactive space.

An aesthetic choice that fits all of these requirements was to make all models yield sequences of musical notes, which is a musical paradigm that is easily recognized by most of the audience. When the sequences are required to be more intense, their notes were increasingly faster and louder. In order to make all musicological models yield sequences that sounded as part of the same piece, they were all bounded to the same octatonic scale, and differences were added on the way each model creates a path within that scale.

As shown in Fig. 4, each generative system is independent from the others. They correspond to four different voices, namely *upper*, *middle*, *harmony* and *bass*. All of them yield note sequences, which will be later rendered.

One sequence generation model applied relies on a Markov chain (*Cope, 1997*), adjusted so that the next note is equally likely to be equal to the previous note, a step down or a step up the scale. This model was used in the *upper* and the *middle* voices, which were also restricted to particular note ranges. The note range restriction allows users to quickly recognize each one of the voices.

The other sequence generation model was the purely random choice. In the *lower* voice, a random note from the scale (within the range restriction) was yielded at each interaction.

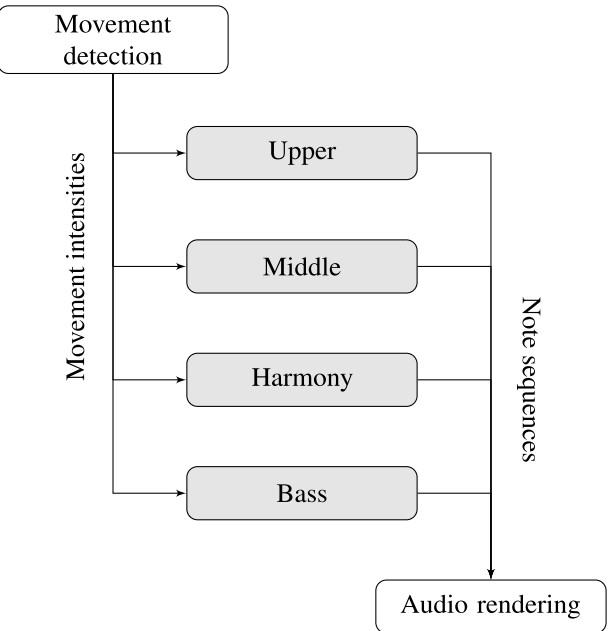

**Figure 4** **Block diagram for the musical interactivity.**

In the *harmony* voice, two random notes from the scale (also within the range restriction) were yielded at each time.

All four voices had different functions to transform the input (movement intensity in the corresponding partition) into values of note speed and loudness, so that more intense movements are mapped to faster and louder notes. These functions were manually adjusted to provide a balanced auditory response related to each space partition, as well as an interesting experience. In all cases, it has proved to be interesting to have a lower bound filtering on the input below which it is considered as noise and does not produce any sonic response.

As a result, MOTUS quickly responds to the audience's actions. It yields a sequence of notes that are almost always dissonant and out of sync related to each other. Nevertheless, the note sequences aim to be perceived as correlated to the audience's movements.

This design fits the employed technology (JavaScript), as it is known for having a bad timing mechanism in current implementations. As note lengths are continuous and not bounded to the notes in other voices, the lack of synchronization does not harm the final result. This also allowed the audio rendering process to be performed by independent agents, as it will be discussed below.

## Audio rendering

The audio rendering process was based on agents that receive pitch, loudness and duration information from the note sequence generated by the musical models. When a note is finished (i.e., its duration has expired), the system proceeds to render the next note (or notes, in the case of the *harmony* voice), and so on. To keep the interactivity process in real

time, the note generation and rendering processes must be synchronized, so that a request for a new note triggers its calculation.

Since this system should be easy to understand, it was chosen that all voices would be rendered as a piano sound, using sampling. This way, it was expected that the majority of the audience would be able to identify the sounds that came from the system, even when bad speakers are used. The rendering system was implemented using a ready-made sequencing library (MIDI.js).

After built, the system was tested both on online and live audiences. This process provided a rich feedback, as will be discussed below.

## AUDIENCE FEEDBACK AND DISCUSSION

Motus was displayed both online and for live audience, which are respectively discussed in 'Online' and 'Live.' These are very different situations, as a live context demands a dedicated space for people to move without harming others, a stronger audio system that is capable of competing with other environmental sounds and a screen that allows visualization from a few meters of distance. This is not the case for online displays, which can be visualized from one's living room or office, thus requiring a less powerful hardware.

### Online

For the online interactions, the system was advertised on social networks, and feedback was obtained both spontaneously and from an optional evaluation form. The questions in the form were:

1. Do you play musical instruments and/or sing? (multiple choice: no, and I am not interested; no, but I would like to; yes, casually; yes, frequently; yes, professionally)

2. Do you dance? (multiple choice: no, and I am not interested; no, but I would like to; yes, casually; yes, frequently; yes, professionally)

3. What audio device did you use in the interaction? (multiple choice: no sound; embedded laptop or desktop audio; small computer speakers; headphones; big audio system or surround)

4. What screen did you use in the interaction? (multiple choice: no screen; computer or laptop screen; big screen or projector)

5. How do you evaluate your interaction with Motus? (multiple choice: not interesting, a little interesting, very interesting, extremely interesting)

6. Describe how was your interaction with Motus (optional, open question)

7. What would you change or add in Motus? (optional, open question)

8. Select from the items below what would you like to do with Motus in the future. (multiple choice, multiple answers: keep interacting; recommend to friends; download as a mobile app; contribute to next version; other)

9. Please, provide any other feedback you find relevant.

In total, 19 volunteer subjects responded the questionnaire, and the majority (16) classified Motus as "very interesting" or "extremely interesting" for question 5. Although

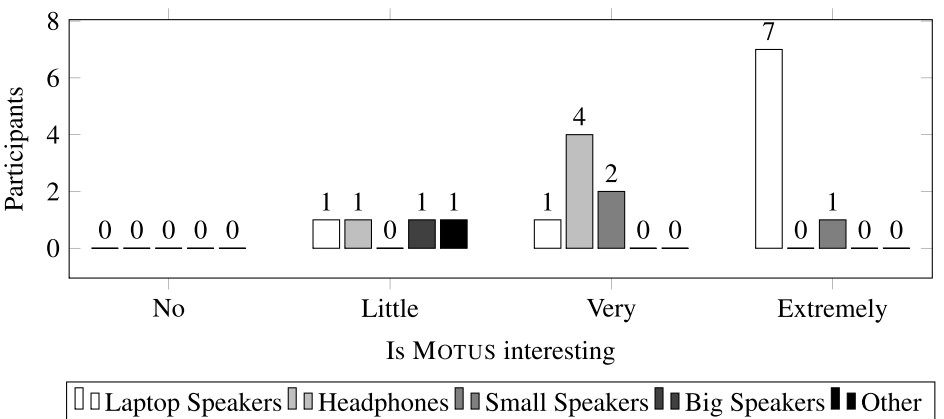

**Figure 5  Interest in MOTUS according to audio hardware.** "Other" refers to one subject that reported using a mobile device for the interaction.

this shows the device was well evaluated, it is also interesting to highlight the conditions that lead to this evaluation. Therefore, these results were jointly analyzed with the answers regarding the hardware used by each subject and their prior interest in dance and music.

As it will be seen, no subject classified MOTUS as "not interesting." This is an encouraging result, but can mean that uninterested subjects simply chose not to answer the questionnaire. Nevertheless, provided answers gave important insight about the audience's usage and perception of the installation.

Figure 5 shows the number of subjects with each type of audio reproduction hardware, grouped by their reported interest in MOTUS (all subjects reported using their default screen for the interaction). It may be noted that using laptop (embedded) speakers did not harm the interaction. On the other hand, no subject using headphones reported MOTUS as "extremely interesting," which can indicate that full body movements are an important part of the experience.

Data indicates that MOTUS was successfully deployed over the web and using ubiquitous hardware, as it was designed for. According to the audience, the use of minimal hardware does not harm the overall experience. However, it is important to detect which aspects impact the subjects' reported interest level.

To detect that, the reported interest levels were grouped according to the subjects' prior interest in dancing or playing instruments and singing, as shown in Fig. 6. Gathered data shows that users with a greater interest in dancing tend to report a greater interest in MOTUS, but a similar behavior is not observed when considering their interest in playing instruments or singing. This is another evidence that performing body movements is related to a more interesting experience with the installation.

All the subjects chose at least one option from Question 8. This shows that possibilities for future use were considered. As shown in Table 1, most subjects would like to keep interacting or recommend to friends, which are indicators of a positive experience with the installation. The answers with "other" regarded using MOTUS in different

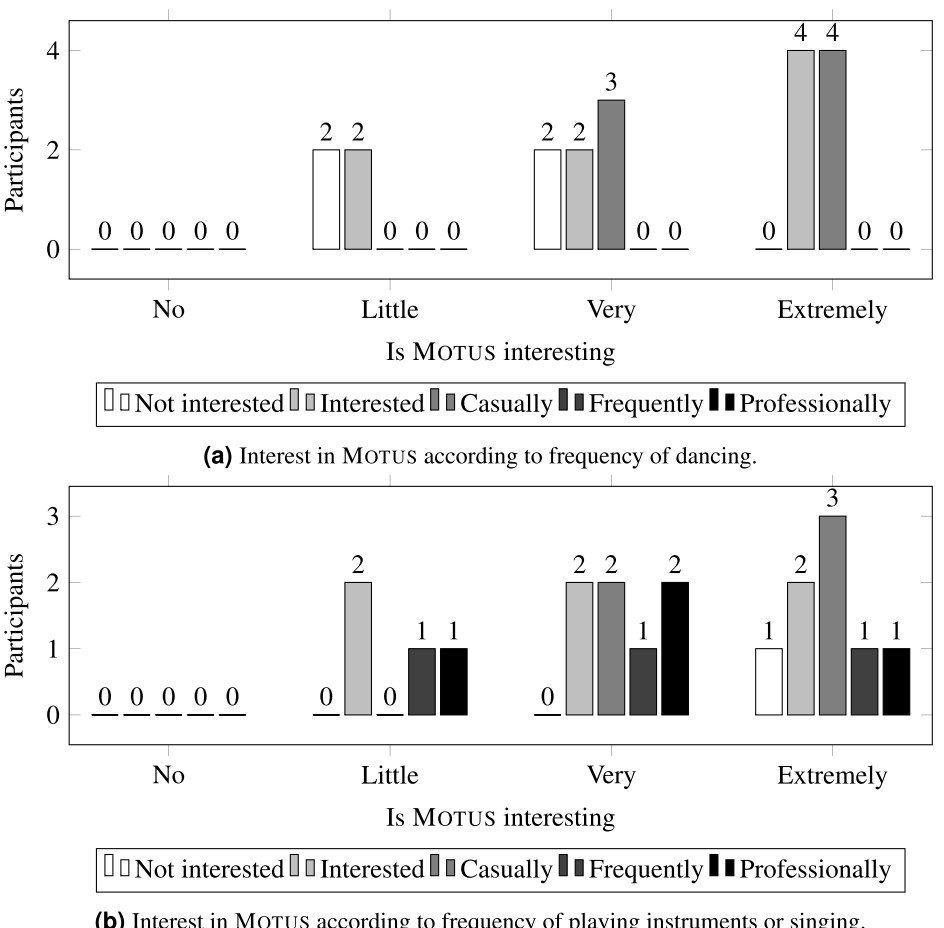

**(a)** Interest in MOTUS according to frequency of dancing.

**(b)** Interest in MOTUS according to frequency of playing instruments or singing.

**Figure 6** Interest in MOTUS according to frequency of artistic activities.

**Table 1 Number of times each option in Question 8 was chosen.**

| Action | Votes |
| --- | --- |
| Keep interacting | 13 |
| Recommend to friends | 13 |
| Download as mobile app | 9 |
| Contribute to next version | 5 |
| Other | 2 |

situations—in a classroom and using a video feed from a landscape—which also points at a positive experience.

The experience description (related to Question 6) showed that most subjects first engaged an exploratory stage, in an attempt to detect the rules governing MOTUS, and then started applying their own repertoire to the interaction. According to the reports, the exploration of own sensations and body movements tended to generate more pleasant

experiences than attempts to generate specific musical movements. The musical generation was perceived as simple, as most subjects were able to quickly understand it.

The majority of the suggestions for future work provided as an answer to Question 7 point toward changing the installation's musicological model. Also, each suggestion was very different from the others, for example: "add more instruments," "I tried to play the blues" and "I don't like the way notes fade out." This is an indication that the musicological model, and probably the screen division for interaction, should be freely composed by users, possibly sharing their results.

Almost all comments regarding the interaction with the webcam related to the division of the screen. Again, each user had a different suggestion, including increasing the number of divisions and constructing neutral areas that could be used to silently move between other areas. Only one comment suggested the use of a finer motion acquisition algorithm, allowing finger positions to be detected.

The spontaneous feedback was obtained by messages sent by e-mail and social networks. Most of it manifested pleasant surprises, as the presence of such a web application was found to be new. They also provided interesting comments regarding the online interaction.

The most common one was a desire to record the interaction and share it on social networks. In an extreme case, a user posted a screenshot online of the visual feedback. This was not implemented in the system, but the demand clearly states a direction for future work.

There was also a demand for porting the system for mobile environments. This is not possible at this moment because of the reduced processing power of mobile devices. However, it wasn't tested how the algorithm would behave if implemented as a native application.

Interestingly, some online users did not allow MOTUS to use their camera, hence harming the experience. This happened because the system was mistaken for privacy-invading malware. A more informative website may be used in the future to prevent this from happening.

## Live

When shown live, MOTUS was mounted using a big screen for video feedback (either a screen or a projector, depending on the venue) and an amplifier for audio playback. It was granted that there would be some free space for movements (as much as possible, which also depended on the venue). Care was taken so that the camera was pointed towards somewhere with no accidental movements (e.g., people passing by or strong shadings from outside) and with a more or less uniform illumination, allowing the camera to work properly.

It was found that the interaction with the system made a small part of the audience too self-conscious to engage in participation. Two possibilities for that are the mirror characteristic of the visual feedback and the tendency of executing random movements in a public space. However, this was not true for everyone.

Part of the audience quickly engaged on exploring the limits within the sonic response of the installation could be controlled. They tended to approach the camera and perform finer arm and hand movements. Some manifested the sensation of playing an imaginary spatial piano.

The more extroverted part of the audience quickly engaged on exploring different body movements. An interesting interaction appeared when groups of people started interacting with the system together, which is perfectly possible due to the nature of the movement detection algorithm. These interactions generally took a longer time and were usually comprised of smiles and laughter.

Finally, an interesting manifestation was given by the audience, especially those who previously played musical instruments. They clearly manifested frustration with the lack of control possibilities in the interaction, as the same movement is not always related to the exact same musical response. Also, there were comments on the simplicity of the interaction process, which made it boring after a few minutes of exploration.

Although ambient lighting is usually a problem in camera-based movement detection systems, it was found that the system is robust to many different conditions. The system worked under different lighting conditions and presented adequate behavior except when lights blinked. However, the interaction experience was slightly changed depending on the colors of the background and the clothes of the audience.

## CONCLUSION

This paper described Motus, a digital interactive audio-visual installation that requires only hardware that is available in most computer and has its software deployed as a web application. The aesthetic concept behind the system is that it should convert movement to music. These premises—the technical deployment and the desired artistic result—led to a series of design and aesthetic decisions, which are thoroughly described.

Motus was shown in live performances, as well as in a website. Feedback was collected from the audience using a questionnaire and by spontaneous comments, which allowed to evaluate how the interaction with the system happened. It was found that this interaction was, most of the times, positive, but sometimes found not very engaging as it doesn't allow many musical aspects to be explored.

It can be accessed at http://www.dca.fee.unicamp.br/~tavares/auxiliary_material/Motus/index.html, and can be freely used by anyone. Currently, it requires the Google Chrome browser. The source code is also available online, at https://github.com/tiagoft/motus.

The installation system is ready to be presented for large audiences, and there seem to be two clear directions for future work. The first is to allow recording of audio and video, as well as sharing of this content in social networks. The second is allow users to compose and share their own interaction models, with a broader range of musical material.

### Funding

This work has been funded by FAPESP, under grant 2013/17329-5. The funders had no role in study design, data collection and analysis, decision to publish, or preparation of the manuscript.

### Grant Disclosures

The following grant information was disclosed by the author:
FAPESP: 2013/17329-5.

### Competing Interests

The author declares there is no competing interests.

### Author Contributions

- Tiago Fernandes Tavares conceived and designed the experiments, performed the experiments, analyzed the data, contributed reagents/materials/analysis tools, wrote the paper, prepared figures and/or tables, performed the computation work, reviewed drafts of the paper.

### Data Deposition

The following information was supplied regarding the deposition of related data:
The source code is available on GitHub (https://github.com/tiagoft/motus).

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
