# Peer review of "An interactive audio-visual installation using ubiquitous hardware and web-based software deployment"

_PeerJ Computer Science, doi:10.7717/peerj-cs.5_

## Round 0.1 · original submission · Major Revisions

The reviewers are unanimous in finding that your submitted work is original, interesting, and within the scope of this journal. However, they also point to several issues that must be addressed:

• the review of earlier research must be extended. Reviewer 1's pointers to relevant conference series and journals should be particularly helpful in updating the literature review.

• the point that Reviewer 2 raises about violating the objective of having all hardware embedded in a single laptop should be addressed.

• both Reviewers 1 and 2 ask for further details on evaluation. If it is not possible to provide a deeper discussion of the evaluation already conducted, then it may be necessary to do further evaluation. This evaluation can be quantitative or qualitative.

* both Reviewers 2 and 3 raise questions about the emphasis on technical aspects of the project over the user experience and the music chosen. Please address these additional aspects to balance out the technical presentation.

We do hope that you will undertake the above revisions. The reviewers are agreed that this paper will be of interest to the research community, that it is an engaging 'read', and that this strand of research is promising.

Reviewer 1 ·

Basic reporting

This paper presents an interactive audio-visual musical installation, MOTUS, developed with low-cost hardware and software and experienced by a possible large audience, e.g., at home.
The topic is interesting and quite original. Analysis of background is based on a few papers and should be extended. For example, I think that a search in journals such as Computer Music Journal or Journal on New Music Research or in the proceedings of conferences such as NIME, SMC, and ICMC would provide useful references for a deeper analysis of the state of the art.
The paper is overall well written, even if some minor language fixes would be needed. It does not comply with the standard sections PeerJ requires, but I think this is acceptable given the peculiarity of the topic, i.e., an interactive audio-visual musical installation rather than a scientific experiment. Figures are relevant, but limited to block diagrams of the system. One or more figures showing the visual output of the installation would be interesting and useful as well. A short video showing the installation at work could be supplied as additional material.

Experimental design

I think the submission is within the scope of the journal: even if this is not a scientific experiment, it can be included in the framework of Sound and Music Computing, which ACM recognized as a specific field of Computer Science. The work is quite original and the techniques used for developing the installation are appropriately described. Nevertheless, I think the approach is quite naïf, grounding on very basic computer vision algorithms and on very basic mappings between movement qualities and sound. While keeping things simple, so that everything can still run on low-cost devices, I think that something more sophisticated and more interesting than basic direct mapping could be achieved.
Whereas, on the one hand, this submission cannot be evaluated using the classical standards as if it were a scientific experiment, on the other hand I think that a deeper evaluation would be worth pursuing. Given the possibly large number of people experiencing the installation, questionnaires for evaluating audience experience with it (e.g., in terms of engagement, pleasantness, interest, and so on) could be administered and a statistical analysis could be performed on them. Such an evaluation would make the paper stronger.

Validity of the findings

See my comments above: given the peculiarity of this submission, it cannot be evaluated with the usual standards for scientific experiments. Nevertheless, a more formal and quantitative evaluation of the installation with the audience would be needed and would provide interesting information about how the installation is perceived.

Additional comments

In conclusion, I think this paper addresses an interesting topic, i.e., making audio-visual musical installations that can be experienced by a large and possibly distributed audience, being the installation grounded on simple techniques running on low-cost devices. However, as it is, the paper is quite weak for journal publication: the approach seems quite naïve and does not sufficiently explore the possibilities that novel technologies open in this field. For example, several different computer vision techniques could be exploited and several different mapping could be applied. Such different versions/installations could be tested (quantitatively) with the audience in order assess e.g., the extent to which they are interesting, engaging, of artistic value.

Reviewer 2 ·

Basic reporting

This paper describes Motus, a low-cost interactive musical installation. I enjoyed reading about this work. The author’s merit is to have evidenced a very important gap in related work. Most of the existing interactive systems for music creation indeed require expensive hardware and software. A very well formulated research objective follows: to develop a musical installation that can be used using a common laptop.

The paper describes in details the algorithms for motion capture and music generation. Despite none of them are particularly innovative for the community, they allow for tracking audience movements and generating the musical output with an acceptable computational burden. With respect to the musical model, I am confused about the decision of the author to "avoid inducing the audience to perform known popular dance moves” and "preventing the induction of expectations on the audience”. Both dancing and fulfilment of expectations are typical conditions of engagement, so it is not clear why they should be avoided. Also, I believe that piano music is not easier to understand than, say, classical music with richer orchestration. A reference to confirm author’s claim is missing.

Experimental design

I found a relevant experimental design flaw. Unexplainably, the set-up of the live audience condition comprises "a big screen for video feedback and an amplifier for audio playback”. This set-up is not aligned with the initial objectives of the work about having all the hardware embedded in a single laptop.

Validity of the findings

Given the objective of the paper, providing an extensive evaluation of the audience experience is necessary to assess the quality of the system. The key contribution of this work would indeed be to provide the audience with an experience that could be comparable, in terms of engagement, to that usually achieved with more expensive installations. Unfortunately, the evaluation methodology is weak, and that’s the biggest limitation of this work. There is no information about the evaluation techniques and about how data were analysed and processed. The author makes use of vague statements such as “it was found that”, missing to reference how these findings came to light. Missing this, the scientific contribution of the paper is highly affected.

Additional comments

The paper should be better ground on the literature. To this end, I suggest the author to improve the related work section given the numerous musical installations about music and movements (Camurri 2013; Morreale 2014) - as well as algorithmic music creation (e.g. the works of Cope, Miranda, Legaspi, ecc).

Finally, pictures of the visual interface and the user interaction would be appreciated.

I like the idea behind Motus and I really encourage the author to keep working with this project, which is very promising. Also, the research objective of opening the access to this form of art to a broader audience is definitely a interesting topic that worth much more investigation. However, the evaluation shortcomings highly impact the quality of the contribution.

·

Basic reporting

The structure of the article seems acceptable. The sequence focus the installation, the movement detection, the video rendering, the music logical models, the audio rendering, the audience feedback and the conclusions. So the structure seems to belong only to a technical communication. My suggestion is to focus better, in the beginning, the cultural field where this experiment was born and the character of the used music, by discovering references as ancient architecture buildings, i.e. Battistero di Pisa.

Experimental design

The search question is clarified and focuses the "easy" way to perform the interactive experiment by using simple devices. The management of the sound output seems to be well performed. The character of the generated music is defined through technical references, parameters and only some cultural references. It's better if the author will try in deep to identify some of these characters also by discovering them inside his own music references.
It's interesting that the generative system is performed with four different voices whose generations are independent each one from the others, by performing a different music logical model. But it's not clarified how they are different and which character will identify each voice, apart from the range of notes.

Validity of the findings

The search is well focused in the technical field and in the musical field. The results are well described. Also if it should be necessary to hear MOTUS for a global evaluation.
My suggestion is to identify the word MOTUS with a peculiar character, being the Latin word used a lot from all the world
So the article seems to be of interest and my suggestion is to publish it with only some revisions, focused in primis on the music culture references.

Additional comments

The author focuses the search of character in the output of sound, clarifying several different parallel events. I suggest to identify better the different characters and the music vision at the base of the different voices and of the ensamble, focusing better the type of control of random factors. So the possibility to increase the peculiar cultural references should be interesting for increasing the quality of the article

---

## Round 0.2 · Minor Revisions

Thank you for so quickly and thoroughly responding to the first round of reviewing!

You have addressed all of the reviewers' suggestions and the paper is very nearly ready to be accepted. As editor, I have only two further modifications to suggest:

• the introduction includes the phrase "a technology-enabled experience that is sensibly different from simply listening to music...". 'Sensibly different' doesn't quite fit here. Perhaps "substantively different" or "perceptively different"?

• In Section 5, it is first stated that 20 people responded to the questionnaire and then a few paragraphs later that 19 people responded. Can you clarify this?

These amendments should be straightforward. If you could send the revised version within the next few days then we may be able to fit your submission into the first edition of the journal.

Congratulations on your successful paper!

---

## Round 0.3 · accepted · Accept

Thank you for so quickly making the final revisions!